# Immunotherapy in Medulloblastoma: Current State of Research, Challenges, and Future Perspectives

**DOI:** 10.3390/cancers13215387

**Published:** 2021-10-27

**Authors:** Marije J. Voskamp, Shuang Li, Kim R. van Daalen, Sandra Crnko, Toine ten Broeke, Niels Bovenschen

**Affiliations:** 1Department of Pathology, University Medical Center Utrecht, 3584 CX Utrecht, The Netherlands; m.j.voskamp@students.uu.nl (M.J.V.); S.Li-2@umcutrecht.nl (S.L.); s.crnko@umcutrecht.nl (S.C.); a.g.tenbroeke-2@umcutrecht.nl (T.t.B.); 2Cardiovascular Epidemiology Unit, Department of Public Health & Primary Care, University of Cambridge, Cambridge CB1 8RN, UK; krv22@cam.ac.uk; 3Center for Translational Immunology, University Medical Center Utrecht, 3584 CX Utrecht, The Netherlands

**Keywords:** medulloblastoma, immunotherapy, immune checkpoint

## Abstract

**Simple Summary:**

Medulloblastoma (MB) is a primary brain tumor and is one of the most prevalent neoplasms in the pediatric age group, with a median age of diagnosis of six years. Besides the symptoms that can accompany BM tumors, conventional therapy involving surgical resection, radiotherapy and chemotherapy can result in severe brain damage and, even with this aggressive, multimodal therapy, the disease is characterized by high relapse rates. As an alternative to the damaging conventional therapies, the use of immunotherapy in MB has gained interest. Different types of immunotherapy, including cancer vaccines, oncolytic viruses, CAR T-cells, and immune checkpoint inhibitors have been studied in in vitro and in vivo models of MB as well as in clinical settings. This review will elaborate on the current state of research concerning immunotherapy in MB, with a special focus on immune checkpoint inhibition and CAR T-cell therapy.

**Abstract:**

Medulloblastoma (MB), a primary tumor of the central nervous system, is among the most prevalent pediatric neoplasms. The median age of diagnosis is six. Conventional therapies include surgical resection of the tumor with subsequent radiation and chemotherapy. However, these therapies often cause severe brain damage, and still, approximately 75% of pediatric patients relapse within a few years. Because the conventional therapies cause such severe damage, especially in the pediatric developing brain, there is an urgent need for better treatment strategies such as immunotherapy, which over the years has gained accumulating interest. Cancer immunotherapy aims to enhance the body’s own immune response to tumors and is already widely used in the clinic, e.g., in the treatment of melanoma and lung cancer. However, little is known about the possible application of immunotherapy in brain cancer. In this review, we will provide an overview of the current consensus on MB classification and the state of in vitro, in vivo, and clinical research concerning immunotherapy in MB. Based on existing evidence, we will especially focus on immune checkpoint inhibition and CAR T-cell therapy. Additionally, we will discuss challenges associated with these immunotherapies and relevant strategies to overcome those.

## 1. Introduction

Primary tumors of the central nervous system (CNS) and brain represent one of the most common neoplasms in the pediatric age group. Among the most prevalent is medulloblastoma (MB), accounting for 8.4% of all primary brain and CNS tumors in the United States among 0–14-year-olds. Compared to adults, the incidence of MB is 10-fold higher in pediatric patients (i.e., 0–21 years of age) [1]. Based on their initial localization and histological characteristics MB tumors are believed to originate from various neuronal stem or progenitor cell populations. They initiate in the posterior fossa region of the brain and commonly disseminate to the leptomeninges or, more rarely, metastasize to locations outside of the CNS (e.g., lungs, bone, or lymph nodes) [2]. In the clinic, MB is subdivided into four groups: Wingless (WNT), Sonic hedgehog (SHH), Group 3 (G3), and Group 4 (G4). These different subgroups are characterized by specific genetic mutations, resulting in different molecular pathways and physiological presentation [3]. Recent research has focused on elucidating the underlying genetic upmake of MB tumors and defining more specific subtypes that could help with better prognosis and treatment response prediction in the clinic [4,5].

The median age of diagnosis of MB is 6, with the majority being diagnosed before 17 [2,3]. MB is characterized by clinical symptoms ranging from mild to severe, including nausea and vomiting, headache, fatigue, ataxia, vision loss, and gait impairments [6]. The identification of MB requires time-consuming diagnostic tools such as histological and molecular characterization, magnetic resonance imaging (MRI), and cytological examination of cerebrospinal fluid (CSF). Consequently, together with heterogeneity in clinical manifestation, usually characterized by symptoms that are easily overlooked, it can take weeks or even months before a diagnosis is confirmed [7]. As a result, tumors are often relatively large and 20–30% of patients exhibit metastasis at diagnosis [8]. Conventional therapy for MB is currently limited and includes surgical resection of the tumor with subsequent radiation and adjuvant chemotherapy [9]. Despite being a multimodal therapy, relapses occur in nearly 75% of pediatric cases within two years [10]. For approximately 30% of patients, relapse eventually results in death. Moreover, conventional treatment strategies often cause severe damage to the brain (e.g., cerebellar mutism syndrome as a risk of surgical resection and reduced white matter integrity associated with neurocognitive deficits due to craniospinal irradiation) [11,12,13]. Considering these limited MB treatment options and the detrimental effect they have on the patient’s quality of life, an urgent need exists for developing better treatment strategies. 

Over the past two decades, immunotherapy has been extensively studied in cancer therapy and numerous immunotherapeutic cancer treatments have entered the clinic [14]. Cancer immunotherapy intends to direct and enhance the body’s own immune system to control and eliminate tumor growth. This type of therapy can be tailored to each patient, making it more specific and, therefore, reducing side effects. This can be achieved through adoptive cell therapy (e.g., using chimeric antigen receptor T (CAR T)-cells, Natural Killer (NK) cells, tumor-infiltrating lymphocytes (TILs)), treatment with immunomodulators (e.g., cytokines), administration of vaccines (e.g., Human papillomavirus (HPV) and Hepatitis B vaccines), or the use of antibodies that trigger the immune system by targeting immune checkpoints (e.g., using ipilimumab [15]). We argue that the molecular difference between the four subgroups of MB is of considerable importance in the immunological TME and response to different types of immunotherapy. In this review, we will provide an overview of the characterization of MB and the current state of research concerning the use of immunotherapy in the treatment of MB, in addition to discussing the challenges and relevant approaches to optimize and personalize these therapies. Based on existing evidence, the main focus will be on immune checkpoint inhibition and CAR T-cell therapy. 

## 2. Medulloblastoma Classification

Over the past decades, research on the pathophysiology of MB has elucidated genetic, epigenetic, and molecular patterns. In 2006, this resulted in an initial proposal to divide MB into two subgroups based on aberrant activation of two developmental cell signaling pathways: Wingless (WNT) and Sonic hedgehog (SHH) [16]. Subsequently, additional research revealed the elaborated genetic characteristics of MB. As a result, in 2012, the classification system was proposed, which is still used in the clinic to date, describing four subgroups: WNT, SHH, Group 3 (G3), and Group 4 (G4) (Figure 1) [3]. More recently, the highly heterogenous nature of MB tumors, even within these subgroups, has been revealed. Specific genetic mutations in MB tumors have been associated with disease prognosis as well as immunological and physiological features of the tumors [4,5]. These characteristics, in turn, correlate with the treatment response to immunotherapy, making a more in depth understanding of the origin and presentation of MB tumors clinically relevant [17,18,19,20,21]. Therefore, the following section is devoted to an overview of MB subtypes and their genetic, histological, and clinical characteristics. 

WNT tumors account for approximately 10% of all MB tumors [22]. These neoplasms originate from dorsal brainstem progenitors and the primary tumors are located in the dorsal brainstem [23]. The majority of WNT tumors are the result of a somatic activating mutation of catenin beta 1 (CTNNB1), leading to a consistent activation of β-catenin and constitutive WNT pathway signaling [4,24]. SSH tumors are responsible for approximately two-thirds of MB cases in infants (<3 years) and adults (>16 years). However, they are less common outside of these age groups, manifesting in only 10–15% of MB patients [22]. Transcriptional profiling, moreover, revealed genetic and transcriptional differences between infant and adult SHH MB patients [37]. SHH tumors are found in the cerebellar hemisphere, where they progress from granule neuron progenitor cells (GNPCs) [25]. In this type, somatic activating mutations of Smoothened (SMO) or deleting mutations of Patched (Ptch) result in independent activation of the SHH pathway, which, in healthy development, regulates many developmental processes with mitosis of GNPCs being just one of them [25,38,39]. SHH tumors are commonly divided into TP53-wildtype and TP53-mutant tumors, as this mutation significantly affects the disease prognosis [40]. With a 5-year survival rate of <60%, G3 MB is considered the most aggressive subgroup. It makes up approximately 25% of all MB diagnoses and is diagnosed twice as often in men [22]. Lastly, approximately 35% of MB patients, with a higher relative prevalence in men, are diagnosed with G4 MB, which is considered moderate compared to the other subgroups [22]. G3 and G4 tumors are located in the superior and inferior parts of the fourth ventricle, respectively [24]. The genetic profiles of the G3 and G4 (non-WNT/non-SHH) tumors are not as well defined. Yet, approximately one-fifth of G3 tumors show amplification of MYC, a transcription factor that is involved in cell cycle progression [4]. G4 tumors are often associated with mutations of chromatin-remodeling genes, such as PRDM6 and KDM6A43 [4]. Likewise, the origin of the tumors is poorly understood. However, it is suggested that both the G3 and G4 tumors may originate from GNPCs through an SSH-independent pathway [26]. It has been further reported that G4 tumors may arise from upper rhombic lip progenitors (URLP) and that their gene expression pattern is similar to cerebellar glutaminergic granule neurons, whereas MYC-associated transformation of astrocyte progenitors has been associated with the development of G3 tumors [26,27,41]. Additional subdivisions of G3 and G4 MD are based on specific genetic alterations and epigenetic factors, but those have not been translated to the clinic to date.

Previous research identified relevant genetic and epigenetic diversity within the four main subgroups [4]. Subsequently, a further subdivision was proposed, involving 12 subtypes based on patterns in genetic mutations and copy number alterations (Figure 1) [5]. Whilst these genes could be exploited as therapeutic targets, the number of mutations is generally relatively low and few therapies targeting these tumor neo-antigens currently exist [42,43]. Besides genetic and epigenetic classifications, the histological character of MB can be divided into three subtypes. Firstly, the classic MB tumors are characterized by moderate pleomorphism of the nuclei, and average amounts of mitotic figures and apoptotic bodies. Secondly, large cell/anaplastic MB tumors consist of cells with remarkable pleomorphism, high amounts of mitotic figures and apoptotic bodies, and hypertrophic cells with single nuclei. Thirdly, the desmoplastic/nodular type MB tumors present confined regions of neuronal cells called nodules, surrounded by layers of connective tissue [28]. Together with the conventional MB type based on the 4-group classification and MB genetic profiles, these histological features play an important part in the risk assessment [5]. Importantly, heterogeneity in genotype, age, tumor histology, and presence of metastases, complexifies the task of offering an accurate MB prognosis and treatment. To tailor future therapies for individual patients, leading to more effective treatment with reduced adverse effects, recognition of the heterogeneous nature of MB and a deeper understanding of its prognostic implications is crucial. An overview of the four main subgroups, subtypes, and a more detailed overview of currently identified genetic factors, histological features, and their associated risk assessment can be found in Figure 1.

## 3. Immunotherapy in Medulloblastoma

Due to the heterogenic nature of MB tumors, immunotherapy has recently been the main focus of novel therapy development. Several immunotherapeutic approaches that aim to increase the immune cell recognition of brain tumors, such as MB, have reached clinical trials, including oncolytic viral therapy and cancer vaccines [44]. Oncolytic viral therapies in cancer, in general, aim to specifically infect and kill tumor cells, achieving this through two processes. Firstly, they kill cancer cells directly by replicating within the cell, causing the cell to burst. Secondly, cell lysis results in the exposure of tumor antigens and subsequent stimulation of a systemic immune response through a process called epitope spreading [45]. However, challenges are to be expected based on previous findings in different types of brain tumors. In glioblastoma, intravascular injection of oncolytic viruses was clinically shown to be ineffective due to inactivation by the host’s immune system and, after orthotropic injection of oncolytic viruses, macrophages and microglia in the TME attacked the virus and prevented its efficacy [46]. Moreover, oncolytic viruses can induce resistant mutation of glioma-associated stem cells, which might cause resistance to, for example, immune checkpoint inhibitors [46,47]. Current phase I trials are evaluating oncolytic viruses based on Cytomegalovirus (NCT03299309 and NCT03615404 (ClinicalTrials.gov, accessed on 25 May 2021)), measles virus (NCT02962167 (ClinicalTrials.gov)), herpes simplex virus-1 (NCT03911388 (ClinicalTrials.gov)) and poliovirus (NCT03043391 (ClinicalTrials.gov)) in the treatment of MB. 

In contrast, cancer vaccines can consist of tumor cells or peptides, and aim to directly target antigen-presenting cells (APCs) that can then present the tumor antigen on MHC molecules [48]. Alternatively, cancer vaccines can be autologous and whole cell based. An example of such a cancer vaccine is the Food and Drug Administration (FDA)-approved sipuleucel-T vaccine that consists of dendritic cells (DCs) activated with a fusion protein antigen that aims to prime the immune system as a treatment of prostate cancer [49]. Other cancer vaccines are DNA- or RNA-based, enabling them to bypass the restricting boundaries of human leukocyte antigen (HLA) haplotypes [48]. This is because the administered template DNA or RNA, once taken up by host cells, is transcribed and translated by the patient’s own cells and this internal processing results in personalized antigens that can be presented to APCs to trigger an immune response [48,50]. However, DNA and RNA vaccines are susceptible to rapid extracellular degradation and can result in an undesired additional immune response, leading to a lack of therapeutic response and possible side effects [51]. Therefore, as often seen in current applications, delivery systems such as autologous DCs are required, which entails an expensive and complex production process. In recent clinical trials focusing on glioblastoma and MB, several cancer vaccines failed to elicit a robust T-cell response, caused severe adverse events, or were ultimately unable to improve the long-term survival rate ([52]; NCT01171469 and NCT02332889 (ClinicalTrials.gov)). A current challenge is that cancer vaccines and oncolytic viral therapies rely on stimulation of the body’s own APC-activated specific T-cell response and migration of these T-cells into the tumor microenvironment (TME), which is limited due to the immunosuppressive nature of most MB tumors. 

Another way to reduce the immunosuppressive nature of the TME is by targeting immune checkpoints (ICs). These ICs are found on immune cells and, upon activation, induce tightly balanced stimulatory or inhibitory signals that regulate immune homeostasis [53]. Interestingly, many cancer types evolve to alter the expression of various inhibitory ICs, enabling them to suppress the local anti-tumor immune response [54]. Therefore, the focus of novel anti-cancer therapies has moved to lifting this suppressive TME. Antibodies that target inhibitory ICs, IC inhibitors (ICI), can reactivate the tumor-specific immune response and, consequently, reduce tumor growth in a broad range of cancer types [55,56]. ICIs have already proven to be beneficial additions to the treatment of multiple cancer types, such as malignant melanoma, non-small cell lung cancer, and triple-negative breast cancer [57]. However, little is known about the possible application of immunotherapy and, more specifically, ICIs in the treatment of MB. The difficulty implicating cancer immunotherapy in MB might partially be explained by the intricate system of physical barriers of the CNS that complicate drug delivery and efficacy [58,59]. Nevertheless, over the past years, evidence suggesting a promising role for ICIs in brain cancer has emerged. Another promising approach to cancer immunotherapy is the use of CAR T-cells that can be made to specifically target surface antigens of tumor cells, which could be a way to overcome this as this type of therapy does not require MHC presentation of the antigens [60]. Therefore, the main focus of this review will be the current state research of concerning immune checkpoint targeting therapies, CAR T-cell therapy, and their associated challenges and future perspective in MB. 

### 3.1. Immune Checkpoint Inhibitors

ICIs have been at the vanguard of the recent cancer immunotherapy revolution, allowing T-cell activation and reactivating an immune response to tumor antigens in numerous types of tumors. To date, the most notable examples are antibodies that block the cytotoxic T-lymphocyte antigen 4 (CTLA-4), and the programmed cell death protein 1 (PD-1) and its ligand PD-L1. Inhibitors of each of these checkpoints have received FDA approval for a still increasing number of malignancies [57]. Additionally, therapies inhibiting newly emerging targets, including mucin domain 3 (TIM-3) and indoleamine 2,3-dioxygenase-1 (IDO1), as well as agonistic therapies that target the positive stimulatory immune checkpoint CD40 are showing promising results in the treatment of brain cancer in preclinical and clinical settings [61,62,63]. 

The efficacy of monoclonal antibodies targeting checkpoints in brain cancer is affected by numerous factors, including molecular tumor characteristics and the accessibility of the TME for immune cell influx. MB tumors are known to have an immunosuppressive and highly heterogenous character, complicating the use of ICI monotherapy compared to more immunologically active tumors. Additionally, due to their highly heterogeneous nature, they are able to develop resistance to ICIs. Therefore, challenges are faced when designing treatment strategies. This section describes, in greater detail, the current state of research concerning the use of several ICIs in MB and the broader prospects for this class of drugs in the improvement of the treatment of MB based on results obtained in other types of brain cancer. Especially for immune checkpoints that are less extensively studied in brain tumors, such as IDO1, TIM-3, and CD40, current knowledge is mainly based on research in other malignant brain tumors, such as glioblastoma. However, since studies often include MB and glioblastoma among the general inclusion of malignant brain tumors, these data are deemed relevant to future research and are believed to indicate possible roles for the immune checkpoints discussed in MB. An overview of the clinical trials involving immune checkpoint-targeting therapies in MB is provided in Table 1.

### 3.2. PD-1/PD-L1 Immune Checkpoints

PD1 (also known as CD279) is a member of the CD28 receptor family; it is expressed on B-cells, activated monocytes, DCs, NK cells, and T-cells, and binds to its ligands PD-L1 or PD-L2 [64]. The PD-1/PD-L1 axis is dysregulated in numerous cancer types and promotes an immunosuppressive milieu through the inhibition of T-cell activation and infiltration. Targeting PD-1/PD-L1 signaling is an attractive strategy for cancer treatment. Several FDA-approved drugs, therefore, target this axis, including the PD-1 inhibitors nivolumab, pembrolizumab, and cemiplimab and the PD-L1 inhibitors atezolizumab, avelumab, and durvalumab [57,65]. However, multiple studies have indicated that the number of infiltrating PD-1^+^ T-cells in the MB microenvironment is limited and that MB tumor cells often show low PD-L1 expression [66,67]. Accordingly, PD-1-inhibiting therapies show discrepant results in MB [68]. However, more recently, studies have been able to identify factors that affect PD-1/PD-L1 blockade efficiency, the optimization of which could improve the treatment efficacy, such as the subtype of MB, the timing of administration, and combination therapies [19,69,70]. 

Firstly, it has been indicated that administration of anti-PD-L1 antibodies 7 days after injection of PD-L1-negative MB tumor cells in mice results in tumor rejection. Administration of anti-PD-L1 antibodies simultaneously with the inoculation of the tumor significantly reduces treatment response [70]. Moreover, this study demonstrated that PD-1^+^ CD8^+^ T-cell infiltration in the tumors treated on day 7 is 4-fold higher compared to isotype and 2-fold higher compared to day 0 treated tumors. These results indicate that the efficacy of PD-L1 blockade in PD-L-negative MB is dependent on timing and the TME at the time of T-cell priming. However, this report did not declare whether post-treatment upregulation of PD-L1 was measured, and the follow-up study mentioned in this report has not yet been published. The negative effect of immediate anti-PD-L1 administration might be due to T-cell exhaustion in which the anti-PD-L1 causes T-cells to have a strong initial proliferation response, followed by a loss of proliferative capacity, making them less effective in the TME long-term. This phenomenon has been described previously in relation to PD-1 or PD-L1 blockade in different tumor types and the importance of T-cell differentiation status has also been indicated in the context of adoptive T-cell therapy [71,72,73]. 

PD-L1 expression and general immune cell influx significantly differ between the different subgroups of MB—as shown by a study on the TME of SHH and G3 MB tumor models in mice [17]. Ptch1-mutated SHH tumors in mice were associated with higher overall numbers of immune cells, such as myeloid derived suppressor cells (MDSCs), tumor-associated macrophages (TAMs), DCs, and infiltrating CD4^+^ and CD8^+^ T-cells, compared to NSC-mutated G3 tumors. The increased immune cell influx in the SHH group is associated with increased expression of inflammation-related genes compared with G3/4 type tumors and especially the TAM-associated genes CD163 and CSF1R, the upregulation of which is suggestive of tumor-promoting M2 type macrophages [74]. Nevertheless, G3 tumors showed a greater response to anti-PD-1 therapy that was reflected by increased survival and resulted from a bigger relative CD8^+^ PD-1^+^ T-cell population, based on which the authors suggested a key role for PD-1^+^ T-cells in NSC MB tumors. Furthermore, in MB patients, high expression of PD-L1 and low levels of infiltrating CD3^+^ or CD8^+^ lymphocytes are predictive of poor prognosis [69]. The lack of anti-PD-1 response in SHH tumors might be due to the increased presence of MDSCs and TAMs that are known to negatively regulate immune responses [17,75]. Besides immunosuppression, TAMs can exert pro-tumor effects through the production of cytokines that are involved in angiogenesis and growth factors that promote tumor progression [76]. Taken together, these results suggest that patients with G3 tumors might benefit from PD-1 targeted therapy, whereas the suppressive TME in SHH tumors indicates these tumors might be more likely to be resistant to PD-1-inhibiting monotherapy. A previous study aimed to identify differences in PD-L1 expression on the different subgroups of MB tumors and found that overall PD-L1 expression is low across all subgroups (in most cases < 1%) but that the SHH subgroup shows the highest degree of PD-L1 expression (>2%) in human MB samples [18]. In the same study, they similarly found higher levels of PD-L1 in SHH than G3/4 cell lines in vitro and, interestingly, that PD-L1 expression could be stimulated by immune response-mimicking exposure of the tumor cells to IFN-γ or radiation in vitro. SHH cell lines show both constitutive and inducible expression of PD-L1, whilst G3/G4 cell lines only show inducible expression. However, this study could only show these differences in vitro and was unable to find an association with immunogenicity of the tumors. In general, these studies are susceptible to varying results due to differences in staining sensitivity and overall low levels of PD-L1 in the tumors, of which a large part might even be expressed by non-tumor cells [17].

To circumvent the limitations of preclinical models and immunohistological profiling, and to increase the sample size, a study aimed to analyze the MB TME using a deconvolution algorithm on a gene expression dataset containing eight different types of brain tumors, including 282 MBs [19]. Again, the results emphasize low overall PD-L1 expression in MB tumors compared to other brain tumors. Moreover, they analyzed the differences between MB subgroups in a dataset containing 763 MBs and found that median PD-L1 expression is highest in WNT tumors, followed by SHH tumors and then G3 and G4 tumors. The subgroups were also characterized by different TMEs. Specifically, SHH tumors are associated with high infiltration of fibroblasts, T-cells, and macrophages, while G3 tumors are characterized by a low number of macrophages but high numbers of CD8^+^ T-cells, and G4 tumors show the lowest number of fibroblasts and the highest number of cytotoxic lymphocytes. Based on the results obtained in this study, the authors proposed a novel method to divide MB tumors into two types of immune evasion. The first mechanism is mediated through macrophages and Treg-cells and is typically found in SHH, WNT, and a subset of G4 tumors. The other mechanism is dependent on anti-inflammatory cytokines and immune checkpoints, mainly characterizing G3 and G4 tumors. This model of subdivision seems to contradict their finding that PD-L1 expression was higher in WNT and SHH tumors compared to G3- and G4-type tumors and the above-mentioned results that suggested PD-1^+^ T-cells were the key factor in immune suppression in G3 tumors in mice. This may partially be explained by the activation of alternative inhibitory checkpoint pathways in G3 and G4 tumors and the fact that gene expression does not necessarily represent the membranous expression of an inhibitory receptor. 

Overall, the studies obtain seemingly discrepant results concerning the association between MB subgroups and PD-L1 expression. The discrepancy in overall expression levels may be explained by differences in study types (i.e., in vitro, in vivo, or ex vivo) and the use of different antibodies and staining conditions as well as varying timings of measurement, resulting in varying degrees of methodological sensitivity and possible failure to detect time-dependent PD-L1 expression. Moreover, recent research shows that expression of PD-1/PD-L1 is not always predictive of the efficacy of PD-1/PD-L1 inhibition. To an extent, PD-1 as well as PD-L1 expression in tumor biopsies was associated with poor prognosis as well as increased survival in different types of cancer [77]. Moreover, significant clinical response to PD-1 inhibition and subsequent increased survival was found in patients with PD-L1^−^ non-small cell lung cancer and this response, in some cases, was superior to that in PD-L1^+^ patients [78]. Furthermore, PD-1 and PD-L1 expression was previously shown to only be predictive of PD-1 inhibition response when measured after initiation of treatment and not when measured pre-treatment, indicating poor predictability of PD-1 inhibitor efficacy [79]. Taken together, these studies emphasize the heterogenicity in MB tumors and the importance of taking the different subgroups and factors such as the timing of administration and immune cell influx into consideration when designing and interpreting studies. Furthermore, recent research into other types of cancer obtained promising results by combining traditional PD-1/PD-L1-inhibiting therapy with other immunotherapeutic approaches, such as CAR T-cells [80] or inhibition of other inhibitory immune checkpoints [81,82]. Further research should reveal the feasibility of these combination therapies in MB. To date, there are four clinical trials that aim to assess the use of PD-1/PD-L1 pathway inhibitors in, among other indications, MB: Pembrolizumab (NCT02359565 (ClinicalTrials.gov)), Nivolumab (NCT03173950 (ClinicalTrials.gov)), Nivolumab in combination with the CTLA-4 inhibitor Ipilimumab (NCT03130959 (ClinicalTrials.gov)), and Nivolumab in combination with the CD122-preferential IL-2 pathway agonist Bempegaldesleukin (NKTR-214) (NCT04730349 (ClinicalTrials.gov)).

### 3.3. B7 Family Immune Checkpoints

The B7 family proteins have been excessively studied for their role in tumor immune evasion. The two major types of B7 proteins are B7-1 (CD80) and B7-2 (CD86), which, upon binding to CD28 on T-cells, induce a stimulatory response and, when binding to CTLA-4, produce an inhibitory signal. The co-inhibitory immune checkpoint B7 homolog 3 (B7-H3, CD276), with an unknown receptor, may also be involved in the immune evasion of brain tumors and, to an extent, MB. B7-H3 has been found to be highly overexpressed in a wide range of solid cancers and its overexpression has been associated with disease severity and poor clinical outcome. However, currently, only two clinical trials that assessed anti-B7-H3 antibodies, namely 8H9 and MGA271, in glioma and other B7-H3-associated cancers, have posted preliminary results that confirm their safety and efficacy [83,84]. 

In MB, B7-H3 was found to be overexpressed in all subgroups [85]. Subsequently, further research confirmed the role of B7-H3 in MYC^+^ MB tumors and the viability of B7-H3 as a target for immunotherapy in MB [20]. Based on this datamining study, it was concluded that expression of B7-H3 is positively correlated with shorter overall survival and that its expression is especially elevated in the more aggressive subgroups G3 and G4. Moreover, in cell lines of MB, they found that MYC promotes B7-H3 expression and that inhibition of MYC could induce downregulation of B7-H3. Further examination revealed a role for miR-29, a micro-RNA known to be involved in tumor suppression [86]. MYC is involved in micro-RNA repression and it was shown to bind the promoter of micro-RNA miR-29, resulting in the downregulation of miR-29 [87]. miR-29 can bind the 3′ untranslated region of B7-H3, resulting in the downregulation of B7-H3 [88]. In MB MYC^+^ MB tumors, the downregulation of miR-29 can, therefore, result in a negative feedback loop between MYC, miR29, and B7-H3 and, in turn, inhibit angiogenesis [20]. 

Another study identified an additional micro-RNA, miRNA-1253, that can exert tumor-suppressive effects in MB via inhibition of B7-H3 [21]. G3 and G4 MB tumors share a common cytogenetic abnormality, i17q. At the end of this isochromosome lies miR-1253, a microRNA that is involved in regulating the expression of bone morphogenic proteins (BMPs) in cerebellar development. miRNA-1253 is hypermethylated in MB and its demethylation results in decreased tumor cell proliferation. Interestingly, B7-H3 was identified as a direct oncogenic target of miRNA-1253; silencing of B7-H3 substantially reduced tumor cell migration and invasion [21]. 

Furthermore, the pathogenic role of B7-H3 might be exerted through its presence in MB cell-derived exosomes [89]. Previous research revealed a role for exosomes in MB immune evasion and, according to this study, B7-H3 overexpression in these vesicles might be part of novel tumorigenic pathways [89,90,91]. Thus, as recent studies emphasize the potential significance of B7-H3 in MB, future research should consider this pathway as a target for immunotherapies in MB. There are two upcoming clinical trials, both studying the effect of B7-H3-targeted radioimmunotherapy in MB (NCT04167618 and NCT04743661 (ClinicalTrials.gov)).

### 3.4. IDO Immune Checkpoints

Indoleamine 2,3-dioxygenase 1 (IDO1) is at the apex of the kynurenine pathway and both IDO1 itself and kynurenine have been linked to tumor immunosuppression [92,93]. As IDO1 is overexpressed in glioma stem cells and as its expression can be induced by IFN-γ stimulation, its involvement in brain tumors and, accordingly, MB has also gained interest [94,95,96]. Furthermore, IDO expression in glioblastoma cells is associated with reduced efficacy of chemo-radiation therapy [97]. Inhibition of IDO significantly increases the ability of chemo-radiation therapy to trigger a complement response and upregulation of VCAM-1 on vascular endothelium at the glioblastoma tumor site. These results indicate that the complement system might be a downstream effector of IDO inhibition. Overall, IDO inhibition could be a relevant strategy in MB by itself, or to improve the efficacy of conventional therapies. Due to the lack of clinical data, however, the possible adverse effects of IDO inhibition in MB are currently unknown. A clinical trial (NCT02502708 (ClinicalTrials.gov)) was recently completed in which IDO-inhibition was combined with chemo-radiation therapy (results yet to be published).

### 3.5. CD40 Immune Checkpoints

CD40, part of the TNF receptor family, is a co-stimulatory receptor involved in the activation of immune responses. CD40 ligation can induce direct cytotoxic effects on CD40-positive tumors, including a large portion of solid tumors [98]. In glioma cells, ligation of CD40 can induce upregulation of vascular endothelial growth factor (VEGF) and shows a correlation with tumor size [99]. Clinically, it was found that expression of CD40/CD40L was positively correlated with disease prognosis in glioblastoma [100]. Furthermore, the beneficial effect of CD40 ligation in combination therapy has been established in mouse models of glioma, in which CD40 ligation improved the efficacy of therapeutic approaches including COX-2 inhibition and several cancer vaccines. These combination therapies resulted in increased CD8^+^ T-cell levels and IFN-γ-production, and reduced Treg-cells in the brain, as well as prolonged survival [100,101,102]. Interestingly, a previous study identified that systemic delivery of CD40 agonist antibodies in a glioma mouse model results in an increase in tertiary lymphoid structures, associated with increased T-cell infiltration, but, remarkably, in response to CD40 agonism, they also observed a dysfunctional cytotoxic T-cell response, upregulation of suppressive CD11b^+^ B-cells, and reduced treatment response to PD-1 inhibitors [103].The above-mentioned results in glioma, together with the low overall infiltration of immune cells in MB and the highly immunosuppressive TME, suggest that combination therapy of ICIs and immunostimulants (e.g.,CD40 ligation) could be a desirable strategy, but further research is needed to elucidate discrepant results. Nevertheless, the ability of CD40 agonistic antibodies to induce hypofunctional and exhausted T-cell populations could result in decreased efficacy of combination therapy, and this should be closely monitored in future clinical trials. Additionally, toxicity of systemic delivery of CD40 agonistic antibodies, such as cytokine-release syndrome, can lead to limited maximum tolerated doses that are insufficient for significant clinical effect [104]. Therefore, local delivery routes might be needed, or synergistic combination therapies could be used in which a lower CD40 dose is required. Based on this preclinical evidence, a new clinical trial is currently recruiting subjects for the assessment of an agonistic CD40 antibody, APX005M, in the treatment of several CNS tumors, including MB (NCT03389802 (ClinicalTrials.gov)).

### 3.6. Tim-3 and Tim-4 Immune Checkpoints

A more recent development in immune checkpoint inhibition is the identification of T-cell immunoglobulin and mucin domain 3 (Tim-3) as a potential target in brain tumors, with current studies mainly focusing on glioma. Tim-3 is overexpressed on tumor-infiltrating lymphocytes in several cancer types and is suggested to be involved in immune evasion and tumor growth [105]. Interestingly, Tim-3 was found to be overexpressed on peripheral CD4⁺ and CD8⁺ T-cells in glioma patients and the level of Tim-3 expression in CD8⁺ T-cells was correlated with tumor grade. Moreover, plasma TNF-α was negatively correlated with Tim-3⁺ CD4⁺ or CD8⁺ T-cells. [106] In addition, the effect of Tim-3 inhibition was evaluated in a mouse model of glioblastoma multiforme, and the results showed that treatment with anti-Tim-3 and anti-Carcinoembryonic Antigen-Related Cell Adhesion Molecule 1 (CEACAM1) combination therapy significantly increases survival time and immune memory against glioma cells. Administration of these antibodies increases CD4^+^ and CD8^+^ T-cell infiltration and decreases Treg proliferation, which are associated with the upregulation of IFN-γ and the downregulation of TGF-β [107]. TIM-4, another receptor in the TIM family, was also associated with immune evasion in glioma and was found to be overexpressed in glioma-derived macrophages [108]. In the same study, glioma-derived T-cells overexpressed phosphatidylserine (PS), a ligand that binds the TIM-4 receptor, and glioma-derived macrophages phagocytosed these PS-expressing T-cells. This, in turn, might induce a tumor-specific Treg population. Based on the promising in vitro and in vivo results, TIM-3 and TIM-4 could be interesting targets in brain oncology and studying their involvement in MB might reveal promising therapeutic strategies, although no clinical trials have been initiated yet. Therefore, at present, no data concerning possible adverse effects of anti-TIM3/4 therapy are available. 

### 3.7. CAR T-Cell Therapy for Medulloblastoma

Adoptive T-cell therapy holds considerable promise for the treatment of MB. However, earlier approaches involving non-specific effector cells, such as NK cells or lymphokine-activated killer (LAK) cells, often failed to show satisfactory efficacy [109,110]. CAR T-cell therapy is a novel, promising type of adoptive cell therapy that is based on patient-derived T-cells and can, therefore, be personally tailored to each patient. CARs are artificial fusion proteins that consist of an extracellular short-chain variable fragment, a hinge-linker region, a transmembrane domain, a costimulatory domain, and an intracellular CD3ζ signaling domain [111]. These receptors are then transfected into autologous or allogeneic peripheral blood T-cells and intravenously administered to the patient. As CAR T-cell therapy does not require a systemic immune response, it is of specific interest for the treatment of MB due to its potential effectiveness in tumors with low mutational load and immune influx [60]. Based on previously established anti-B7-H3 antibodies, B7-H3 CAR T-cells were tested in mouse models of several pediatric cancers, including MB. B7-H3 was highly expressed on MB tumor cell lines and treatment with B7-H3 CAR T-cells in a xenograft model of MB resulted in significant and specific anti-tumor effects and tumor regression [112]. Furthermore, HER2-targeted CAR T-cells containing a CD3ζ signaling domain and a 4-1BB costimulatory motif (BBz) show robust anti-tumor activity against MB cell lines in vitro and in MB mouse models, where they result in rapid tumor regression [113]. Additionally, a different type of HER2-CAR T-cells was already proven to be clinically effective in glioblastoma [114]. Similarly, CAR T-cells that specifically target Preferentially Expressed Antigen in Melanoma (PRAME), an intracellular protein whose peptide is presented on HLA-A*02:01 receptors for T-cell recognition and that is expressed in 82% of MB patient biopsies, may represent a promising innovative approach for treating patients with HLA-A*02^+^ based on its initial efficiency found in vitro and in an orthotopic mouse model of MB [115]. Several clinical trials are being performed using CAR T-cells in pediatric brain tumors. These include the ongoing phase I trial NCT04185038 (ClinicalTrials.gov), assessing the use of B7-H3-specific CAR T-cells in, among others, MB. Another trial, NCT04270461 (ClinicalTrials.gov), aimed to continue in the drug pipeline of NKG2D-based CAR T-cells, that had previously shown cytotoxicity against NKG2DL^+^ cell lines in vitro and against NKG2DL^+^ cell xenografts in vivo. The trial included patients with MB as well as other types of tumors but was unfortunately withdrawn without reported results. Thus, at present, our knowledge concerning CAR T-cell therapy for MB is still limited to in vitro and in vivo experiments.

As T-cells can penetrate the blood–brain barrier (BBB) and infiltrate the brain in a diffuse manner, a successful tumor-targeted T-cell therapy in MB would obviate the challenges posed by poor drug delivery to the tumor [66]. Furthermore, MB progenitor cells possess unique molecular signatures and reside within protective niches in the tumor. CD133, Nestin and Musashi [116], as well as development-related genes such as Ebfs [117] can be overexpressed by MB stem cells and may act as potential tumor-associated antigen targets for CAR T-cells in MB [118]. Nevertheless, CAR T-cell therapy still has to overcome certain critical challenges such as antigenic escape, in which tumors downregulate the target antigen, resulting in treatment resistance. This has been reported previously in CAR T-cell therapy in leukemia and can be especially expected in MB because of the heterogenicity of these tumors [119,120]. However, multivalent CAR T-cells can be designed to overcome antigenic escape and interpatient variability [121]. Furthermore, the development of antigen-specific CAR T-cells is still a time-consuming, expensive process. Moreover, severe neurotoxicity events such as cytokine release syndrome have occurred in response to CAR T-cell therapy against CD19 in multiple clinical trials, as opposed to the absence of severe adverse events in antibody-based treatments [120,122,123]. As a solution, future studies should investigate the feasibility of CAR T-cells that are activated only upon the binding of multiple tumor-associated antigens to improve specificity to the tumor cells and reduce off-target binding. Alternatively, related therapies are underway, such as CAR-engineered NK cells, which are being studied in numerous tumor types, including glioblastoma [124,125].

## 4. Challenges of Immunotherapy in Medulloblastoma

Although the idea that the brain is one of the immune-privileged sites (i.e., that antigens do not trigger an immune response) has become controversial, the BBB excludes the vast majority of cancer therapeutics from entering the brain parenchyma and mAbs are subjected to rapid elimination from the brain parenchyma by a mechanism called reverse transcytosis [126]. Whereas CAR T-cells and small peptides (such as those used in cancer vaccines) can readily pass the BBB, larger molecules such as the ICIs Nivolumab and Ipilimumab are usually prevented from entering the CNS [127,128,129]. However, with the progression of brain tumors, angiogenesis, and gradual impairment of the BBB, the immunosuppressive character of the TME will likely become the main obstacle to drug efficacy. The MB TME is associated with factors that complicate drug delivery and efficacy, such as hypoxia, low extracellular pH, and high interstitial fluid pressure (IFP). [28]. The latter can increase due to leaky tumor vasculature where, as the IFP increases, it flows into surrounding tissues, preventing drug passage into the tumor tissue [130]. Moreover, MB tumors can develop a neovasculature, which facilitates tumor growth, survival, and metastasis, and also affects drug delivery [131]. This neovasculature, in some cases, possesses BBB-like characteristics. Remarkably, it was demonstrated that the neovascular architecture of MB tumors differs between subgroups in mouse models [132]. This study showed that murine WNT-MB tumors maintain a highly dense, non-BBB-like vasculature with fenestrated endothelium, allowing for a better therapeutic response. In contrast, murine SHH-, G3- and G4-MB tumors show a BBB-like neovasculature, with non-fenestrated endothelium and low vascular density. The latter results in less treatable tumors. Confirming this hypothesis, they found that while IgG leaked into the tissue fluid of WNT tumors, no IgG was found in SHH tumors [132].

Furthermore, MB tumors do not have a highly immunogenic environment. This is partially explained by their mutational load, which is slightly above average compared to other pediatric malignancies; however, when compared to adult malignancies, this burden is still relatively low. Negative feedback mechanisms further exist in the TME. For example, when CD8^+^ CTLs become activated and start producing IFN-γ, this induces PD-L1 upregulation on tumor cells [133]. Moreover, the distinct immunologic profiles of the different subtypes of MB affect the treatment response to immunotherapy, as explained above in relation to PD-1 targeting. 

To overcome the challenges of immunotherapy in MB, strategies have been identified that improve the efficacy of ICI or to overcome resistance to ICI. For example, TNF can enhance the efficacy of anti-PD-1 antibodies in p53-mutant MB in mice [134]. Alternatively, radiation has been successfully used to enhance the efficacy of immunotherapy in lung malignancies [135,136]. Interestingly, patients with melanoma brain metastasis show an increased treatment response to anti-PD-1 and anti-CTLA-4 with concurrent stereotactic radiosurgery (SRS) (i.e., immunotherapy administered within 4 weeks prior to or after SRS) compared to non-concurrent treatment. This is reflected by a significantly greater reduction in lesion volume and a trend towards increased overall survival, although the latter was not a primary outcome of this study and was not statistically significant [137,138]. It was hypothesized that radiotherapy may have pro-inflammatory effects through the release of damage-associated molecular patterns (DAMPs) and toll-like receptors, and increased expression of MHC-1, and also that CD8^+^ T-cells are involved in the anti-tumor effect of tumor irradiation [139]. Furthermore, the combination of multiple checkpoint inhibitors has proven successful in previous studies. Combination therapy with PD-1 and CTLA-4 inhibitors elicits an anti-tumor response in murine models of SHH and G3 MB [17]. Furthermore, anti-PD-1/anti-CTLA-4 combination therapy has shown efficacy in advanced or metastatic renal cell carcinoma in patients with PD-L1 expression levels below 1%, indicating that the low overall PD-L1 expression in MB tumors might not be predictive for the response to combination treatment [140].

## 5. Conclusions

In summary, previous clinical, in vivo, and in vitro studies targeting immune checkpoints for the treatment of MB have resulted in conflicting findings. The efficacy of therapeutic approaches of IC inhibition varies significantly between studies and discrepant findings have been obtained concerning general characteristics such as expression levels of immune checkpoints and the immunologic character of MB tumors. However, it is noteworthy that: (i) MB subtypes were often not considered whilst being significantly different in terms of TME, molecular signature, and immunosuppressive mechanisms; (ii) study sample sizes were often small; (iii) most preclinical studies were performed in mice, which have different immune functioning compared to humans; and (iv) mouse models mirror the physiology of grown individuals and not of pediatric MB patients. As mentioned previously, the genetic characteristics of MB subtypes are associated with differential expression of immune checkpoints on the tumor cells, as well as with differences in immune cell influx and neovasculature. Furthermore, while animal models of MB are often not representative of the human, pediatric physiology and immune response, clinical trials face certain limitations as well. For example, most clinical trials assess different types of brain tumors in one study, resulting in the failure to consider the different MB subtypes. Additionally, since MB is mainly a pediatric disease and immunotherapy should be tailored to their situation, clinical testing should be conducted on children, complicating the recruiting process. However, recent research gave rise to novel approaches to improving ICI delivery and efficacy, thereby raising new interest in IC targeting. Promising studies assessing antibodies targeting ICs such as PD-1/PD-L1, CTLA-4, B7-H3, TIM-3/4, and IDO1, as well as combination therapies, are now moving into the clinical phase. There are still many ICs identified in different types of tumors that have not yet been evaluated in MB and could be interesting future targets. Additionally, finding ways to boost the immune response, for example, through pro-inflammatory cytokine stimulation or the activation of stimulatory co-receptors, might hold future potential. Furthermore, CAR T-cells that circumvent most of these challenges and specifically target personally tailored antigens present a viable strategy.

We believe that current studies often fail to consider the large differences in the molecular profile of subtypes of MB tumors and, thus, their potential differential treatment response. This understanding of the heterogeneity of the disease and how the genetic mutations found in each subgroup affect the immunological TME, we believe, might be a key factor in the future development of immunotherapies in MB. In future studies, the heterogenicity of MB, as well as factors affecting treatment efficacy, such as the timing of administration and synergistic combinations, should be better taken into consideration in study designs and treatment approaches. With the recent novel approaches and better insight into the specific subtypes within the established 4-group subgroup division of MB, we are moving towards more specific, personalized immunotherapy that can significantly improve patient quality of life compared to conventional treatment.

## Figures and Tables

**Figure 1 cancers-13-05387-f001:**
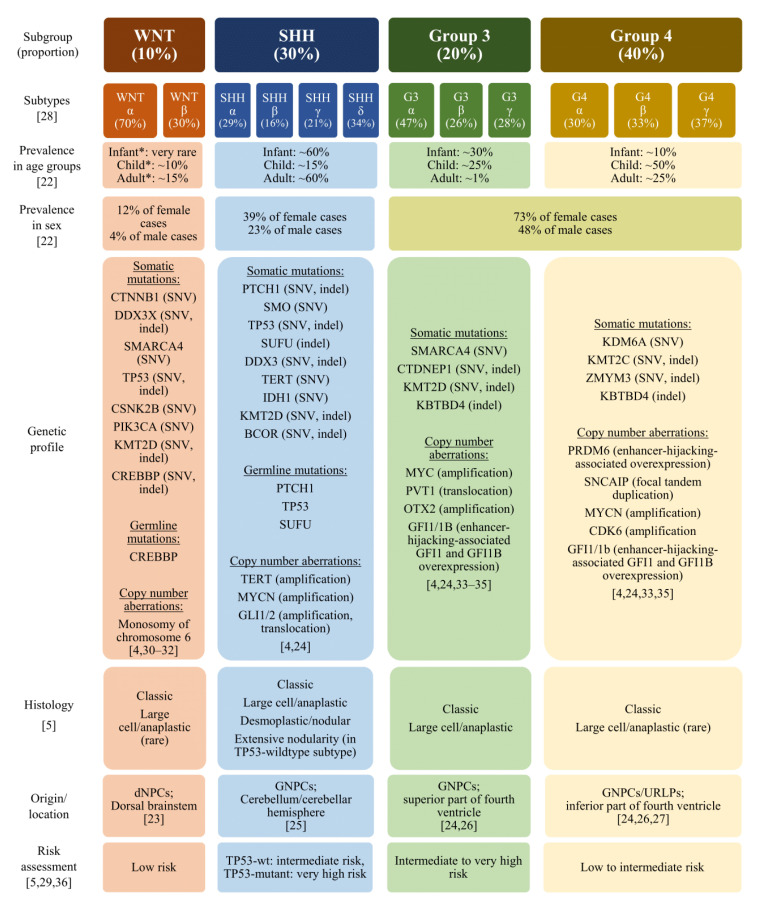
Characteristics of the different subtypes of medulloblastoma. Abbreviations: WNT, wingless-related integration site; SHH, Sonic Hedgehog; dPNCs, differentiating neural stem and progenitor cells; GNPCs, granule neuron precursor cells; URLPs, upper rhombic lip progenitors. * Risk assessment varies depending on genetic subtype. Figure generated by author based on existing literature [4,5,22,23,24,25,26,27,28,29,30,31,32,33,34,35,36].

**Table 1 cancers-13-05387-t001:** Clinical trials involving immune checkpoint-targeting immunotherapy in medulloblastoma.

Trial ID	Title	Phase	Treatment	Target	Indications	Age	N	Status
**NCT** **04730349**	A Study of Bempegaldesleukin (BEMPEG: NKTR-214) in Combination with Nivolumab in Children, Adolescents and Young Adults with Recurrent or Treatment-resistant Cancer (PIVOT IO 020)	1/2	i.v. nivolumab with bempegaldesleukin (BEMPEG: NKTR-214)	PD1, CD122	Ependymoma Ewing sarcoma High-grade glioma Leukemia and lymphoma Medulloblastoma Miscellaneous brain tumors Miscellaneous solid tumors Neuroblastoma Relapsed, refractory malignant neoplasms Rhabdomyosarcoma	<18 and 18–30 years	228	Not yet recruiting
**NCT** **03130959**	An Investigational Immuno-therapy Study of Nivolumab Monotherapy and Nivolumab in Combination with Ipilimumab in Pediatric Patients with High Grade Primary CNS Malignancies (CheckMate 908)	2	Nivolumab, ipilimumab	PD1, CTLA-4	Various Advanced Cancer (including MB)	6 months–21 years	166	Active, not recruiting
**NCT** **03173950**	Immune Checkpoint Inhibitor Nivolumab in People with Recurrent Select Rare CNS Cancers	2	i.v. nivolumab	PD1	Medulloblastoma Ependymoma Pineal region tumors Choroid plexus tumors Atypical/malignant meningioma	>18 years	180	Recruiting
**NCT** **02359565**	Pembrolizumab in Treating Younger Patients with Recurrent, Progressive, or Refractory High-Grade Gliomas, Diffuse Intrinsic Pontine Gliomas, Hypermutated Brain Tumors, Ependymoma or Medulloblastoma	1	i.v. pembrolizumab	PD1	Constitutional Mismatch repair Deficiency syndrome Lynch syndrome Malignant glioma Recurrent brain neoplasm Recurrent childhood ependymoma Recurrent diffuse intrinsic pontine glioma Recurrent medulloblastoma Refractory brain neoplasm Refractory diffuse intrinsic pontine glioma Refractory ependymoma Refractory medulloblastoma	1–29 years	110	Recruiting
**NCT** **03838042**	INFORM2 Study Uses Nivolumab and Entinostat in Children and Adolescents with High-risk Refractory Malignancies (INFORM2 NivEnt)	1/2	Nivolumab and entinostat	PD1	CNS Tumor Solid Tumor	6–21 Years	128	Recruiting
**NCT** **02502708**	Study of the IDO Pathway Inhibitor, Indoximod, and Temozolomide for Pediatric Patients with Progressive Primary Malignant Brain Tumors	1	Oral indoximod with radiation therapy, temozolomide, or cyclophosphamide and etoposide	IDO	Glioblastoma multiforme Glioma Gliosarcoma Malignant brain tumor Ependymoma Medulloblastoma Diffuse intrinsic pontine glioma Primary CNS tumor	3–21 years	81	Completed
**NCT** **04049669**	Pediatric Trial of Indoximod With Chemotherapy and Radiation for Relapsed Brain Tumors or Newly Diagnosed DIPG	2	Oral indoximod with combinations of temozolomide, cyclophosphamide, etoposide, lomustine and radiation therapy.	IDO	Glioblastoma Medulloblastoma Ependymoma Diffuse intrinsic pontine glioma	3–21 years	140	Recruiting
**NCT** **03389802**	Phase I Study of APX005M in Pediatric CNS Tumors	1	i.v. APX005M	CD40	Glioblastoma Multiforme High-grade astrocytoma, NOS CNS primary tumor, NOS Ependymoma, NOS Diffuse intrinsic pontine gliomas Medulloblastoma	1–21 years	45	Recruiting
**NCT** **04167618**	177Lu-DTPA-Omburtamab Radio-immunotherapy for Recurrent or Refractory Medulloblastoma	1/2	177Lu-DTPA-omburtamab radio-immunotherapy	B7-H3	Pediatric medulloblastoma	3–19 years	40	Not yet recruiting
**NCT** **04743661**	131I-Omburtamab, in Recurrent Medulloblastoma and Ependymoma	2	cRIT 131I-omburtamab radio-immunotherapy with Irinotecan, temozolomide, and bevacizumab.	B7-H3	Recurrent medulloblastoma Recurrent ependymoma	<22 years	62	Not yet recruiting

Abbreviations: i.v., intravenous; IDO, indoleamine-pyrrole 2,3-dioxygenase; PD1, programmed cell death protein 1. All data concerning the clinical trials were obtained from ClinicalTrials.gov (accessed on 25 May 2021).

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
