# Peer review of "Immunotherapy in Medulloblastoma: Current State of Research, Challenges, and Future Perspectives"

_cancers, 2021, doi:10.3390/cancers13215387_

Round 1
Reviewer 1 Report
This is a review by a group of epidemiologists, pathologists and immunologists that targets the issue of immunotherapy in medulloblastoma. The review is interesting, though some aspects are overemphasized, and others are not discussed in sufficient detail (see below). This should be addressed in a revised version.
In addition and as a major comment, the authors postulate that the subtypes of medulloblastoma may be distinctly amenable to immunotherapies, and this sounds plausible. However, the hypothesis should be more clearly depicted, as it is of clinical impact and may shape the further development of immunotherapies in medulloblastoma.
Another major comment would be that the review should more precisely focus on te reasons why immunotherapies in medulloblastoma have mostly failed up to now and by what means this could be improved.
Simple abstract / abstract: it is overemphasized that medulloblastoma is commonly metastasized at time of diagnosis. Please adapt. Survival rates are around 65%, depending of the subtype, which is not as bad as in other brain tumors. Please adapt. Immunotherapy has evolved on the basis of a development in different cancers, and not as much because medulloblastoma is such a deadly disease. Sparing toxicity might be another good argument, specifically in children.
Introduction: damage to the frontal lobe white matter is not a specific complication in medulloblastoma. Frontal lobe dysfunction is based on destruction of cerebellar-frontal connections within the brain.
Classification: it is basically 4 subgroups with 2 groups within SHH, namely SHH-dependent/p53 wild type and SHH-dependent/p53 mutated. Not sure if the refined description of the molecular subtypes is necessary in this review. There is no obvious connection between this description and immunologic therapies in text and table. Please describe such connections or shorten paragraph. The same is true for the (though nicely drafted) figure 1.
Oncolytic viruses: the description is more general, and some mechanistic details strengthening the rationale should be included. In addition, the failure of current clinical trials should be more thoroughly discussed.
CAR-T cells: the paragraph is very general, and no specific connection to medulloblastoma is made. If there are no specific data, the paragraph would be better placed in an outlook section. In every case, possible chances and threats should be discussed. The topic comes back later in more detail, and here, again the focus should be on medulloblastoma. Then both parts should be connected.
Immune checkpoint inhibitors: the introduction is too general. The list of trials is not complete, e.g. the trial of INFORM2 with Nivolumab and Etinostat is missing (NCT03838042). The rest of the immune checkpoint section is extensive and could be focused in some aspects. Much of the cited data is not on medulloblastoma, and I would suggest to keep medulloblastoma always in the focus of this review.
Drug delivery: then authors here connect their review to abscopal effect of that can be observed in radiotherapy / immunotherapy connections. This should be discussed more clearly.
Author Response
Dear editor and reviewers,
Thank you for insightful comments on our paper. We have been able to incorporate changes to reflect the suggestions provided by the reviewers and highlighted the changes within the manuscript. Below you can find a point-to-point response to the reviewer’s comments and concerns.
The revisions have significantly improved the manuscript and we are looking forward to receiving your response. Thank you again for your time and feedback on our work.
Kindest regards.
Reviewer #1
This is a review by a group of epidemiologists, pathologists and immunologists that targets the issue of immunotherapy in medulloblastoma. The review is interesting, though some aspects are overemphasized, and others are not discussed in sufficient detail (see below). This should be addressed in a revised version.
In addition and as a major comment, the authors postulate that the subtypes of medulloblastoma may be distinctly amenable to immunotherapies, and this sounds plausible. However, the hypothesis should be more clearly depicted, as it is of clinical impact and may shape the further development of immunotherapies in medulloblastoma.
Response: We thank the reviewer for this comment and have more clearly emphasized this hypothesis in the introduction (line 82-84) as well as in the conclusion (line 615-619).
Another major comment would be that the review should more precisely focus on the reasons why immunotherapies in medulloblastoma have mostly failed up to now and by what means this could be improved.
Response: We agree with the reviewers and have accordingly added sections to further explain the limitations of the different approaches mentioned in the article. The added sections can be found in the following lines: oncolytic viruses, line 171-175; DNA/RNA cancer vaccines, line 192-197; immune checkpoint inhibitors, line 238-242; PD-1/PD-L1, line 279-285; CD40, line 443-450.
- Simple abstract / abstract: it is overemphasized that medulloblastoma is commonly metastasized at time of diagnosis. Please adapt. Survival rates are around 65%, depending of the subtype, which is not as bad as in other brain tumors. Please adapt. Immunotherapy has evolved on the basis of a development in different cancers, and not as much because medulloblastoma is such a deadly disease. Sparing toxicity might be another good argument, specifically in children.
Response: We have removed the statements about metastasis at the time of diagnosis and the survival rates in the simple summary and abstract. Instead, we emphasized the need for novel therapies because of the toxicity associated with conventional therapies (line 26-28).
- Introduction: damage to the frontal lobe white matter is not a specific complication in medulloblastoma. Frontal lobe dysfunction is based on destruction of cerebellar-frontal connections within the brain.
Response: We thank the reviewer for this comment and have adjusted this section accordingly (line 69-70).
- Classification: it is basically 4 subgroups with 2 groups within SHH, namely SHH-dependent/p53 wild type and SHH-dependent/p53 mutated. Not sure if the refined description of the molecular subtypes is necessary in this review. There is no obvious connection between this description and immunologic therapies in text and table. Please describe such connections or shorten paragraph. The same is true for the (though nicely drafted) figure 1.
Response: We agree with the reviewers that the important role of the genetic profile of the MB subtypes in treatment response to immunotherapy was not sufficiently clear. Besides the previously mentioned added emphasis in the introduction and conclusion, we have added a rationale for the detailed description (line 97-103).
- Oncolytic viruses: the description is more general, and some mechanistic details strengthening the rationale should be included. In addition, the failure of current clinical trials should be more thoroughly discussed.
Response: We have added a more clear explanation of the mechanism by which oncolytic viral therapies work (line 166-170) as well as a section about their limitations and why they have often fail to elicit a clinical response (line 171-175).
- CAR-T cells: the paragraph is very general, and no specific connection to medulloblastoma is made. If there are no specific data, the paragraph would be better placed in an outlook section. In every case, possible chances and threats should be discussed. The topic comes back later in more detail, and here, again the focus should be on medulloblastoma. Then both parts should be connected.
Response: We believe that the introductory sentence about CAR T-cells seemed misplaced in the section and moved this sentence to further down the introduction to immunotherapy in MB to make it clearer that CAR T-cells will be discussed later in the article.
We agree that the focus should be on MB, which is why almost all data referenced in the later section about CAR T-cells is on MB and that studies concerning different contexts are only referenced when we feel the data is important for future approaches in MB research.
- Immune checkpoint inhibitors: the introduction is too general. The list of trials is not complete, e.g. the trial of INFORM2 with Nivolumab and Etinostat is missing (NCT03838042). The rest of the immune checkpoint section is extensive and could be focused in some aspects. Much of the cited data is not on medulloblastoma, and I would suggest to keep medulloblastoma always in the focus of this review.
Response: We have added a section to describe in more detail what challenges are faced in ICI therapy in MB (line 238-242). We thank the reviewers for pointing out the clinical trial that was not mentioned in our overview and have added trial NCT03838042 to the table.
We agree that, when possible, data should be focused on MB and for the more extensively studied ICs, this is the case in our review. However, for less extensively studied in brain tumors, such as IDO1, TIM-3, and CD40, current data is mainly based on other malignant brain tumors. We believe that these results are of interest for MB and have added a section to explain our justification for including of this data (line 245-250).
- Drug delivery: then authors here connect their review to abscopal effect of that can be observed in radiotherapy / immunotherapy connections. This should be discussed more clearly.
Response: We have added a more in-depth explanation of this effect to this section (line 570-579).
Reviewer 2 Report
Immunotherapy in medulloblastoma: current state of research, challenges, and future perspectives
1-Please correct BM tu‐ line 14
2-Based on their initial localization and histological characteristics MB tumors are believed to originate from various neuronal stem or progenitor cell populations. For instance, astrocyte progenitors?
3- As in G3, the prevalence of G4 is higher among men. Please reword this sentence since the way it is written is misleading.
4- In the section regarding oncolytic viral therapies please include a few sentences about the potential off-target effect of oncolytic virus therapies. I would advise mentioning all side-/ off-target effects for therapies.
5- This is because the administered template DNA or RNA, once taken up by host cells, is transcribed and translated by the patient’s own cells, and this internal processing results in personalized antigens that can be presented to APCs to trigger an immune response. Please mention the downside of using DNA/RNA.
6-The use of CAR T‐cells that can be made to specifically target surface antigens of tumor cells could be a way to overcome this as this type of therapy does not require MHC presentation of the antigen. A very brief description, consider expanding.
7-Please provide the downside of ICs therapies, line 186.
8- I appreciate that the authors have covered the general issue with any CNS targeting therapy (BBB) but it would be good to dissect this barrier in light of the therapies mentioned.
9- As mentioned before the limitations/ side effects or downside of the checkpoint inhibitors could be mentioned.
10- Firstly, it has been indicated that administration of anti‐PD‐L1 antibodies 7 days after 231 injections of PD‐L1 negative MB tumor cells in mice results in tumor rejection. Administration of anti‐PD‐L1 antibodies simultaneously with the inoculation of the tumor significantly reduces treatment response. Please explain the reason.
11- SHH tumors in mice were associated with higher overall numbers of immune cells, such as MDSCs, tumor‐associated macrophages (TAMs), DCs, and infiltrating CD4+ and CD8+ T‐cells, compared to G3 tumors. What is the reason?
12- Nevertheless, G3 tumors showed a better response to anti‐PD‐1 therapy, resulting from a bigger relative CD8+ PD‐1+ T‐cell population. Greater? Does this correlate with a more favourable prognosis?
13- What may be the reason that PD-1 is higher in G3 compared to SHH while PD-L1 is lower?
14- The PD-1/L1 levels in the 4 subgroups of MB can be displayed in a table/ figure.
15- Moreover, in cell lines of MB, they found that MYC promotes B7‐H3 expression and that inhibition of MYC resulted in downregulation of B7‐H3. What is the molecular link between B7-H3, miR-29, and MYC?
16- The downside of IDO1, CD40, and TIM3/4 should be mentioned.
17-What is the limitation of CAR T cells specifically in MB? Apart from neurotoxicity.
18- A possible solution is the design of CAR T‐cells that target multiple antigens or combination therapies using two types of CAR T‐cells each targeting a different antigen. Could this increase the risk of off-target effects too?
19- Please also discuss the immune response of CNS in comparison with the rest of the body.
20- The MB TME is associated with factors that complicate drug delivery and efficacy, such as hypoxia, high interstitial fluid pressure, and low extracellular pH [30]. Moreover, MB tumors can develop a neovasculature, which facilitates tumor growth, survival, and metastasis, and also affects drug delivery. How does the increase in the intracranial pressure due to the tumour play into this equation?
21- Alternatively, radiation has been used successfully to enhance the efficacy of immunotherapy in lung malignancies [118,119]. Interestingly, patients with melanoma brain metastasis show an increased response to immune checkpoint therapy when treated within 1 month after radiation therapy. What is the justification?
22- What is the link between MB subtype and TME characteristics? What is the anatomical/ physiological basis?
23- Perhaps a mention of clinical trial limitations and concerns in children.
Author Response
Dear editor and reviewers,
Thank you for insightful comments on our paper. We have been able to incorporate changes to reflect the suggestions provided by the reviewers and highlighted the changes within the manuscript. Below you can find a point-to-point response to the reviewer’s comments and concerns.
The revisions have significantly improved the manuscript and we are looking forward to receiving your response. Thank you again for your time and feedback on our work.
Kindest regards.
Reviewer #2
1-Please correct BM tu‐ line 14
Response: We have changed the settings in the document to ‘Don’t Hyphenate’.
2-Based on their initial localization and histological characteristics MB tumors are believed to originate from various neuronal stem or progenitor cell populations. For instance, astrocyte progenitors?
Response: We have found literature that describes a possible role of astrocyte progenitors in the development of G3 type MB tumors and have accordingly added this to the MB classification section (line 132-134).
3- As in G3, the prevalence of G4 is higher among men. Please reword this sentence since the way it is written is misleading.
Response: We have rewritten this sentence to improve clarity (line 121-123).
4- In the section regarding oncolytic viral therapies please include a few sentences about the potential off-target effect of oncolytic virus therapies. I would advise mentioning all side-/ off-target effects for therapies.
Response: We thank the reviewers for this general advise and agree that this was lacking in several sections. With regard to the oncolytic viruses, we have added a section about the pitfalls and challenges of this type of therapy (line 171-177).
5- This is because the administered template DNA or RNA, once taken up by host cells, is transcribed and translated by the patient’s own cells, and this internal processing results in personalized antigens that can be presented to APCs to trigger an immune response. Please mention the downside of using DNA/RNA.
Response: We have added a section describing the downside of DNA/RNA vaccines (line 192-197).
6-The use of CAR T‐cells that can be made to specifically target surface antigens of tumor cells could be a way to overcome this as this type of therapy does not require MHC presentation of the antigen. A very brief description, consider expanding.
Response: We believe that the introductory sentence about CAR T-cells was misplaced in the section and moved this sentence to further down the introduction to immunotherapy in MB to make it clearer that CAR T-cells will be discussed later on in the article (line 219-222).
7-Please provide the downside of ICs therapies, line 186.
Response: We have added a section describing the general challenges of ICIs (line 238-242).
8- I appreciate that the authors have covered the general issue with any CNS targeting therapy (BBB) but it would be good to dissect this barrier in light of the therapies mentioned.
Response: We agree that the connection to the previously mentioned therapies was missing and have added a connecting section (line 539-541).
9- As mentioned before the limitations/ side effects or downside of the checkpoint inhibitors could be mentioned.
Response: We thank the reviewer for this suggestion and have added a general section about the challenges of ICI therapy in MB (see point 7) as well as separate sections about the currently known downsides of specific checkpoint inhibitors. The latter will be described in more detail in point 16.
10- Firstly, it has been indicated that administration of anti‐PD‐L1 antibodies 7 days after 231 injections of PD‐L1 negative MB tumor cells in mice results in tumor rejection. Administration of anti‐PD‐L1 antibodies simultaneously with the inoculation of the tumor significantly reduces treatment response. Please explain the reason.
Response: A section has been added to give a more in-depth explanation as to why the administration of PD-L1 antibodies simultaneously with tumor inoculation might result in reduced efficacy (line 279-285).
11- SHH tumors in mice were associated with higher overall numbers of immune cells, such as MDSCs, tumor‐associated macrophages (TAMs), DCs, and infiltrating CD4+ and CD8+ T‐cells, compared to G3 tumors. What is the reason?
Response: We thank the reviewer for this comment and believe that a more in-depth analysis of the referenced data was in place here. We have accordingly added a section discussing these results (line 291-294).
12- Nevertheless, G3 tumors showed a better response to anti‐PD‐1 therapy, resulting from a bigger relative CD8+ PD‐1+ T‐cell population. Greater? Does this correlate with a more favourable prognosis?
Response: Indeed, the G3 tumors showed a greater response to anti-PD-1 therapy and this was associated with increased survival. We have added this to the section in question (line 295-298).
13- What may be the reason that PD-1 is higher in G3 compared to SHH while PD-L1 is lower?
Response: The relatively increased presence of MDSCs and TAMs in SHH tumors might result in a negatively regulated immune response which might explain this finding. This explanation has been added to the section in question (line 299-301).
14- The PD-1/L1 levels in the 4 subgroups of MB can be displayed in a table/ figure.
Response: Although we like the idea of a clear overview of the PD-1/L1 levels in the 4 subgroups of MB, we believe that, as is emphasized in the text as well, literature on this is limited and studies obtain discrepant results. Therefore, we believe that, at this moment, a comprehensive conclusion can not yet been drawn and a table might be misleading.
15- Moreover, in cell lines of MB, they found that MYC promotes B7‐H3 expression and that inhibition of MYC resulted in downregulation of B7‐H3. What is the molecular link between B7-H3, miR-29, and MYC?
Response: We thank the reviewer for this comment and have added a clearer description of the connection between MYC, miR-29, and B7-H3 in MB tumors (line 386-391).
16- The downside of IDO1, CD40, and TIM3/4 should be mentioned.
Response: As mentioned above, we agree that this information was lacking in the sections pointed out by the reviewers. We have added a section about the downsides of CD40 antagonism (line 443-450). However, for IDO1 and TIM3/4 therapies currently no specific adverse effects could be found. Therefore, we have stated this more clearly in the sections about IDO1 (line 419-420) and TIM3/4 (line 476-477).
17-What is the limitation of CAR T cells specifically in MB? Apart from neurotoxicity.
Response: In the section about CAR T-cells, we described both the neurotoxicity as well as the risk of antigenic escape, especially in MB tumors because of their heterogenic nature. We have adjusted the sentence about the use of multivalent CAR T-cells to overcome antigenic escape and interpatient variability to more clearly emphasize these issues (line 523-524). Furthermore, we have added the time-consuming and expensive production process as a downside of CAR T-cells (line 525-526).
18- A possible solution is the design of CAR T‐cells that target multiple antigens or combination therapies using two types of CAR T‐cells each targeting a different antigen. Could this increase the risk of off-target effects too?
Response: We thank the reviewers for this comment and agree that the latter solution was irrelevant to the context and have rewritten this sentence to more clearly emphasize the potential of CAR T-cells that are activated by multiple antigens only, as those would only increase specificity (line 529-531).
19- Please also discuss the immune response of CNS in comparison with the rest of the body.
Response: In different sections throughout the article, we describe the specific changes in the immunological environment of MB tumors and the overall reduced immune response in the CNS compared to tumor environments elsewhere in the body. However, generally speaking, the immune response in the CNS is similar to the systemic immune response.
20- The MB TME is associated with factors that complicate drug delivery and efficacy, such as hypoxia, high interstitial fluid pressure, and low extracellular pH [30]. Moreover, MB tumors can develop a neovasculature, which facilitates tumor growth, survival, and metastasis, and also affects drug delivery. How does the increase in the intracranial pressure due to the tumour play into this equation?
Response: We have clarified the role of high interstitial fluid pressure in drug delivery in MB tumors (line 544-547).
21- Alternatively, radiation has been used successfully to enhance the efficacy of immunotherapy in lung malignancies [118,119]. Interestingly, patients with melanoma brain metastasis show an increased response to immune checkpoint therapy when treated within 1 month after radiation therapy. What is the justification?
Response: We agree with the reviewer that the justification for this statement was missing and have added a better explanation of the data mentioned here (line 570-579).
22- What is the link between MB subtype and TME characteristics? What is the anatomical/ physiological basis?
Response: In the immunotherapy section, several characteristics of the immunological response and physiological characteristics such as neovasculature are linked to the different MB subtypes. To make this more clear, we have added a summarizing statement in the conclusion section (line 597-599).
23- Perhaps a mention of clinical trial limitations and concerns in children.
Response: We thank the reviewers for this suggestion and have added a statement about this to the concluding paragraph (line 599-605).
Round 2
Reviewer 2 Report
The authors have addressed my comments